# Gypsum, crop rotation, and cover crop impacts on soil organic carbon and biological dynamics in rainfed transitional no-till corn-soybean systems

Khandakar R. Islam[1]*, Warren A. Dick[2], Dexter B. Watts[3], Javier M. Gonzalez[4], Norman R. Fausey[5], Dennis C. Flanagan[4], Randall C. Reeder[6], Tara T. VanToai[5], Marvin T. Batte[7]

1 Soil, Water, and Bioenergy Resources, The Ohio State University South Centers, Piketon, Ohio, United States of America, 2 School of Environment and Natural Resources, The Ohio State University, Wooster, Ohio, United States of America, 3 National Soil Dynamics Laboratory, USDA-ARS, Auburn, Alabama, United States of America, 4 National Soil Erosion Research Laboratory, USDA-ARS, West Lafayette, Indiana, United States of America, 5 Soil Drainage Research Unit, USDA-ARS, Columbus, Ohio, United States of America, 6 Food, Agricultural and Biological Engineering, The Ohio State University, Columbus, Ohio, United States of America, 7 Agricultural, Environmental and Developmental Economics, The Ohio State University, Columbus, Ohio, United States of America

☯ These authors contributed equally to this work.
* islam.27@osu.edu

**Data Availability Statement:** The data and statistical analyses have included in supporting material.

## Abstract

Soil organic carbon (SOC), a core soil quality indicator, is influenced by management practices. The objective of our 2012–2016 study was to elucidate the impact of gypsum, crop rotation, and cover crop on SOC and several of its biological indicators under no-till in Alabama (Shorter), Indiana (Farmland), and Ohio (Hoytville and Piketon) in the USA. A randomized complete block design in factorial arrangement with gypsum (at 0, 1.1, and 2.2 Mg/ha annually), rye (*Secale cereal* L.) vs no cover crop, and rotation (continuous soybean [*Glycine max* (L) Merr., SS] vs corn [*Zea mays*, L.]-soybean, both the CS and SC phases) was conducted. Composite soils were collected (0–15 cm and 15–30 cm) in 2016 to analyze microbial biomass C (SMBC), SOC, total N, active C, cold and hot-water extractable C, C and N pool indices (CPI and NPI), and C management index (CMI). Results varied for main effects of gypsum, crop rotation, and cover crop on SOC pools, total N, and SOC lability within and across the sites. Gypsum at 2.2 Mg/ha increased SMBC within sites and by 41% averaged across sites. Likewise, gypsum increased SMBC:SOC, active C, and hot-water C (as indicators of labile SOC) averaged across sites. CS rotation increased SOC, active C, CPI, and CMI compared to SS, but decreased SMBC and SMBC:SOC within and across sites. CPI had a significant relationship with NPI across all sites ($R^2$ = 0.90). Management sensitive SOC pools that responded to the combined gypsum (2.2 Mg/ha), crop rotation (CS), and cover crop (rye) were SMBC, SMBC:SOC, active C, and CMI via SMBC. These variables can provide an early indication of management-induced changes in SOC storage and its lability. Our results show that when SOC accumulates, its

**Funding:** Rafiq Islam, Project # 1520-732-7226; United Soybean Board; https://www.unitedsoybean.org/ The funder had no role in study design, data collection and analysis, decision to publish, or preparation of the manuscript.

**Competing interests:** No authors have competing interests.

lability has decreased, presumably because the SMBC has processed all readily available C into a less labile form.

## Introduction

Soil organic carbon (SOC) consists of diverse organic compounds that affect soil and environmental functions [1–4]. It is one of the core indicators of soil quality, and acts as a reservoir of substrates, energy, enzymes, and nutrients that maintains biodiversity and efficiency, regulates biochemical reactions and buffering, and provides soil physical stability [5–7]. However, considering SOC as one homogeneous pool ignores variations in the chemical composition and relative proportion of its individual pools, and how these SOC pools both influence and reflect agricultural management practices [8].

The various SOC pools are diverse in terms of their chemical composition, lability and biochemical turnover, and physico-chemical stability [1, 9, 10]. Because a detectable change in the SOC often takes several years, researchers have, therefore, emphasized the need to measure labile SOC pools, thus providing early indications of management-induced changes in SOC content associated with soil quality [5, 8, 11].

Labile C is defined as a small pool of SOC that has rapid turnover rates and is preferably biochemically utilized compared to the rest of the SOC [7, 8, 12]. The SOC pools that are considered as labile are active C, SMBC, potentially mineralizable C, light-fraction, cold and hot-water extractable C, anthrone reactive C, particulate organic C, and beta-glucosidase activity [5, 11–14]. Several metabolic quotients and indices such as basal respiration, SMBC:SOC, microbial cell death rates, specific maintenance respiration rates, and C and N management indices (CMI and NMI) have been suggested as sensitive indicators of SOC and TN dynamics [4, 14–16]. While the amount of C associated with SMBC is small, it is considered a key labile biological component that catalyzes SOC and N transformations. Hence, it is considered of prime importance for labile SOC utilization and nutrient cycling in soil [4, 17].

Management practices of repeated annual plowing, unbalanced and excessive chemical fertilization, and continuous monocropping have led to rapid depletion of labile SOC pools. Early detection of temporal changes from labile to non-labile SOC pools, or vice-versa, caused by agricultural management practices can be useful in helping producers change their practices before they become difficult to reverse and remediate. Agricultural management practices that diversify and maintain a balance between SOC accumulation and lability can improve soil quality and support economic crop productivity. An adoption of sustainable agricultural practices (i.e., continuous no-till, crop rotation with cover crops, residue management, precision fertilization, and soil amendments) is expected to improve soil's functional capacity and lead to enhanced agroecosystem services [7, 16].

Recently, gypsum, which is hydrated calcium sulfate ($CaSO_4 \bullet 2H_2O$) mineral from mined geologic deposits, a co-product of the phosphate fertilizer industries, or a byproduct of flue gas desulfurization of coal-based power plants, has received increased attention as an alternate source of Ca and S for plant nutrition, a proactive chemical amendment for environmental remediation, and for soil quality improvement [16, 18–21]. Gypsum is widely used as a chemical amendment to reclaim sodic and marginal soils for crop production, a conditioner of soil physical properties to reduce surface crusts and improve soil structures, an electrolyte to improve water infiltration, and a treatment to reduce edge-of-field loss of soluble reactive phosphorus [22, 23]. Recently, a few studies have reported that gypsum application favors soil biology and enzymatic activity in Hagerstown silt loam (pH 6.2) in Pennsylvania, USA [16]

and in alkaline-saline clay soils in northwest China [24]. Gypsum amendments reportedly provide several nutrients essential for crop growth in Marvyn loamy sand in Alabama (Shorter), Blount silt loam in Indiana (Farmland), Hoytville clay in northwestern Ohio, and Omulga silt loam in southern Ohio, respectively [21].

Gypsum can play a critical role in regulating SOC transformations due to its high concentration of Ca [18, 21]. The cationic bridging effects of Ca promote flocculation of clay and SOC in the formation of water-stable aggregates that lead to the physical accumulation of SOC as particulate organic C [16, 18]. In contrast, several other studies report that Ca-based amendments, such as lime and gypsum, may lead to SOC depletion due to increased activity of opportunistic microbes via a priming effect on native SOC content [24]. While research on gypsum amendments has primarily focused on improving or restoring physical and chemical properties of sodic and marginal soils, more information is needed to elucidate its effects on soil biology and SOC pools under agronomic cropping diversity.

The cropping diversity that currently dominates the Midwest United States is the corn-soybean (CS) rotation. The success of annually plowed soil that often accompanies the CS rotation has not come without environmental costs and has resulted in SOC depletion over time [25]. Crop rotation, when combined with conservation tillage, especially no-till, is a biological diversity management option to improve soil functionality [25, 26]. It provides several agroecosystem services such as diverse ground covers to shorten fallow periods, minimize soil erosion, suppress weed and disease infestations, improve hydraulic conductivity, and increase soil aggregate formation with an associated decrease in soil crusting, sealing, and compaction [26–28].

Likewise, cover crops, when included into crop rotations, provide extended ground cover, support rhizosphere effects, and exert synergistic effects on SMBC and biological activities, thereby promoting root growth that improves soil aeration and drainage. Cover crops also serve as a source of diverse and biochemically labile C and essential nutrients, phytochemicals, and extracellular enzymes, leading to efficient nutrient recycling and increased SOC accumulation [29]. In recent years, there has been increased interest in integrating cereal rye as a winter hardy cover crop within CS rotations [28–30]. Rye quickly resumes growth in the early spring from winter dormancy and produces a substantial amount of biomass across a broad range of site conditions. By adding surface and subsurface biomass inputs, rye aids in soil aggregate formation that leads to physical protection of SOC, improves soil quality, and increases crop productivity resilience to climate change events [16, 31].

Considering the close relationships among SOC and soil biological, chemical, and physical properties and processes [4, 8, 16], information on agricultural management-induced effects related to early changes in SOC accumulation and lability is important to evaluate soil quality. However, long-term effects of gypsum, crop rotation, and cover crop on soil biology, total N, and SOC pools, under a rainfed no-till system across different sites and weather conditions, but under similar management practices, remain mostly unexplored.

We hypothesize that the integration of gypsum in CS rotations with rye as a winter cover crop will provide diverse organic inputs that create favorable site conditions for increased biological efficiency, and thereby potentially influence the SOC pools across geographic locations. A five-year field research study was conducted under diverse soil and climatic conditions of the Midwest and Southeastern United States with an objective to evaluate how gypsum and cereal rye under different phases of CS rotation impact soil biology, total N, and SOC pools in a rainfed, transitional no-till system.

## Materials and methods

### Site description

No-till field experiments were conducted simultaneously at four different sites under rainfed conditions across different soil types and climate zones (Fig 1) that included Alabama (Agricultural Experiment Station's E.V. Smith Research Center-Field Crops Unit, 32˚25'19" N, 85˚53'7" W at Shorter), Indiana (Davis Purdue Agricultural Center near Muncie (40˚15'34.7" N

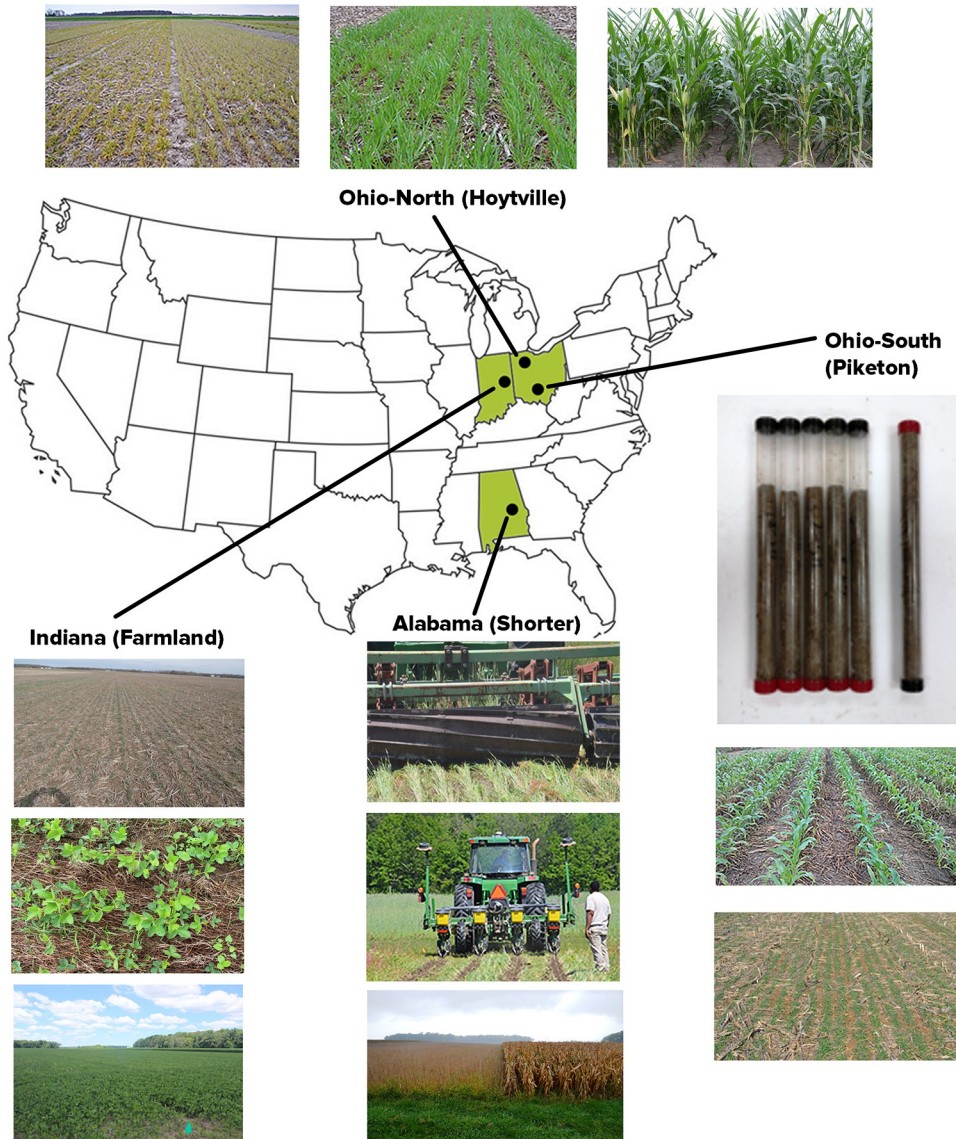

**Fig 1. Gypsum, crop rotation and cover crop rainfed field experiments at Alabama (pictures depict crimper rolling of rye winter cover crop followed by plating of corn and soybeans, and matured corn-soybeans before harvesting), Indiana (pictures depict rye as a winter cover crop followed by growing of soybeans within rye mulch, and soybeans at maximum vegetative growth stage), Ohio-north (pictures depict rye as a winter cover crop in early and late spring before planting of corn, and corn at early maturity stage), and Ohio-south (pictures depict rye as a winter cover crop followed by growing of corn within rye residues, and collected soil cores in plastic tubes using the JMC® stainless-steel soil environmental probe; https://www.jmcsoil.com) in the U.S. Midwest during 2012 to 2016 (Photos taken by Randall C. Reeder and Khandakar R. Islam).**

85˚09'19.9" W at Farmland), and Ohio (Northwest Agricultural Research Station, Ohio Agricultural Research & Development Center, 41˚13'18.26" N, 83˚45'34.91" W at Hoytville and Ohio State University South Centers research farm, 39˚06'78" N, 83˚00'92" W at Piketon) in the U.S. Midwest [21, 32, 33].

Soil at the Alabama site (Shorter) is a deep, well-drained, and moderately permeable Marvyn loamy sand (fine-loamy, kaolinitic, thermic Typic Kanhapludult). Initial analysis showed soil at the 0–15 cm and 15–30 cm depths, respectively, had values of pH 5.9 and 5.5, electrical conductivity (ECe) 170 and 101 μS/cm, SOC 0.39 and 0.30%, total N 0.032 and 0.028%, active C 111 and 79 mg/kg, extractable Ca 0.39 and 0.32 g/kg, extractable sulfur 0.50 and 0.56 g/kg, and bulk density (ρb) 1.53 and 1.62 g/cm.

In contrast, soil at the Indiana site (Farmland) is an association of poorly drained Blount silt loam (fine, illitic, mesic Aeric Epiaqualfs), Glynwood silt loam (fine, illitic, mesic Aquic Hapludalfs), and Pewamo silty clay loam (fine, mixed, active, mesic Fluvaquentic Endoaquolls). Soil at the 0–15 cm and 15–30 cm depths, respectively, had values of pH 6.0 and 6.1, ECe 290 and 173 μS/cm, SOC 1.22 and 0.97%, total N 0.105 and 0.090%, active C 347 and 294 mg/kg, extractable Ca 1.76 and 1.45 g/kg, extractable sulfur 0.38 and 0.40 g/kg, and ρb 1.25 and 1.29 g/cm$^3$.

Soil at the Ohio north site (Hoytville) is a poorly drained Hoytville clay (fine, illitic, mesic Mollic Epiaqualfs). Soil at the 0–15 cm and 15–30 cm depths, respectively, had values of pH 6.2 and 6.3, ECe 315 and 278 μS/cm, SOC 1.93 and 1.82%, total N 0.213 and 0.193%, active C 407 and 303 mg/kg, extractable Ca 4.32 and 4.72 g/kg, extractable sulfur 0.41 and 0.41 g/kg, and ρb 1.32 and 1.39 g/cm$^3$. In contrast to the Ohio north site, soil at Ohio south site is a moderately well-drained Omulga silt loam (fine-silty, mixed, active, mesic Oxyaquic Fragiudalfs). Soil at the 0–15 cm and 15–30 cm depths, respectively, at this site had values pH 6.1 and 5.9, ECe 88 and 66 μS/cm, SOC 0.59 and 0.33%, total N 0.060 and 0.032%, active C 262 and 107 mg/kg, extractable Ca 1.05 and 0.86 g/kg, extractable sulfur 0.20 and 0.15 g/kg, and ρb 1.41 and 1.62 g/cm$^3$ [21].

## Experimental design and cultural practices

Prior to establishing the experiment, all sites were under some type of tillage operations and a CS rotation without any cover crops or gypsum amendments. A three-factorial experiment (3 gypsum rates x 2 cover crops x 3 crop rotations) in a randomized complete block design, with four replications for each treatment combination, was established at all locations in the autumn of 2012 [21, 33]. The gypsum rates were 0, 1.1, and 2.2 Mg/ha, which were annually surface-applied to the transitioning no-till, except at the Ohio south site, where gypsum was applied only in the fall of 2012 and 2014. Crop rotation included continuous SS versus both phases of CS rotation each year with cereal rye as a cover crop versus no cover crop. Cereal rye at the rate of 60 kg/ha was drilled into soil after harvesting of corn or soybeans in October during the experimental period.

Agronomic practices required by the experimental design were managed commensurate with local weather, soil, and pest and weed pressure conditions [21, 32]. All management practices, with the exception of the treatments applied, were performed as similar as possible across all treatments in regard to the geographic, climatic, and time scales. Furthermore, agronomic cultural practices and cover crop planting were performed accordingly. Cover crop was terminated with crimper roller and Roundup® approximately two weeks before soybean or corn planting (Fig 1).

## Soil sampling and analysis

Soil cores (2.54 cm internal dia.) were extracted using the JMC® stainless-steel soil environmental probe lining with plastic tube immediately after crop harvest in October 2016 from

geo-referenced sites within each replicated plot at 0–15 cm and 15–30 cm depth intervals. A minimum of three field-moist soil cores were composited for each replicated plot and placed in sealable plastic bags for temporary storage at 4°C until needed. A portion of the field-moist soil was gently sieved through a 2-mm mesh prior for measuring SMBC. Another portion of the field-moist soil was air-dried for a period of approximately 15 days under shade at room temperature, processed, and analyzed for chemical and physical properties [21].

The SMBC was measured by a rapid microwave irradiation and extraction method [34] with the determination of total extracted organic C in an extract aliquot using a Shimadzu® total dissolved SOC/TN analyzer. The SMBC was calculated as follows,

$$\text{SMBC (mg/kg)} = (C_{MW} - C_{UMW})KME^{-1}$$

where the $C_{MW}$ and $C_{UMW}$ represent extracted C in microwaved and unmicrowaved soils, and the KME is the fraction of the SMBC (0.213) extracted by 0.5M $K_2SO_4$.

The metabolic quotient (SMBC:SOC) was calculated by dividing the SMBC by the SOC [15]. The SOC and total N were determined on finely ground (< 125 μm), air-dried soil by the dry combustion method using a FlashEA-1112 series CNHS-O analyzer®. Active C was determined spectrophotometrically on a separate soil sample after a mild potassium permanganate oxidation (0.02 M at pH 7) of air-dried soil [8]. Both cold and hot-water C were also extracted on separate soil samples and determined using a Shimadzu® total dissolved SOC/TN analyzer [11, 16].

Basic soil chemical and physical properties such as pH (1:2 in soil and distilled deionized water suspension) by glass electrode, ECe (1: 1 soil and distilled deionized water paste) by electrical conductivity probe, and ρb by standard core methods were determined.

## Soil organic carbon and nitrogen lability and management indices

Using the measured SOC, active C, SMBC, and cold and hot-water C data, the C management index (CMI) was calculated as follows [13, 16]:

$$\text{CMI} = (\text{CPI x CLI}) \tag{1}$$

where the CPI is the C pool index and the CLI is the C lability index, which were calculated as:

$$\text{CPI} = [\text{SOC in the treatment soil/SOC in the control soil}] \tag{2}$$

$$\text{CLI} = [\text{CL in the treatment soil/CL in the control soil}] \tag{3}$$

where CL refers to the lability of C, which was calculated as:

$$\text{CL} = [\text{Labile C/Non} - \text{labile C}] \tag{4}$$

Using the same principle, the nitrogen pool index (NPI) was calculated as follows [16]:

$$\text{NPI} = [\text{TN in treatment soil/TN in the control soil}] \tag{5}$$

The labile C pool was considered as a portion of the SOC pool that was comprised of active C, SMBC, or cold and hot-water C. The non-labile C pool was calculated by subtracting the active C, SMBC, or cold and hot-water C pool from the SOC [16].

## Statistical analysis

Multivariate statistical analyses were performed by using SAS® on the measured and calculated soil properties attributed to the impacts of gypsum application, crop rotation, cover crop,

and their interactive effects for each experimental site [35]. While gypsum, crop rotation, and cover crop were considered as fixed variables, block was considered as a random variable to perform analysis of variance (ANOVA). Moreover, to evaluate the impacts of gypsum, crop rotation, and cover crop across all the replicated research sites, a meta-analysis was performed [35, 36]. To conduct a meta-analysis of the pooled data from all sites, first data were normalized by dividing the treatment values of a variable with the control value of that same variable at each site to provide a ratio. The controls were the zero rate of gypsum application, the CS rotation, and the minus cover crop treatment. The normalized ratios were then statistically analyzed using the ANOVA procedures as described above to evaluate gypsum, crop rotation, and cover crop main effects, and their interaction effects on SMBC, SOC pools, TN, CPI, and CMI indices across all sites. For all statistical analyses, the significant main effect and interactions of treatment variables on soil properties were separated by the F-protected Least Significant Difference (LSD) test with a value of $p \leq 0.05$ unless otherwise mentioned. Regression and correlation analyses were performed using SigmaPlot$^{\circledR}$.

## Results and discussion

Significant but variable main effects of gypsum application, crop rotation, and cover crop on soil properties such as SMBC, SOC pools, total N, and SOC lability indices without consistent interactive effects among and across the sites were observed (Tables 1–5). Accordingly, the main effects of gypsum application, crop rotation, and cover crop on soil properties are discussed (Tables 1–5).

### Gypsum impacts on soil carbon pools and lability

While the SOC and total N were unaffected, the labile SOC pools were variably influenced by gypsum application rates (Tables 1–5). In this section, all comparisons are made in reference to the 0-gypsum rate treatment unless noted otherwise. The SMBC, as a biological component of the labile SOC pool, was increased with gypsum treatment at all sites (25 to 33%) with the

**Table 1. Gypsum and cover crop impacts on total soil organic C (SOC), total nitrogen (TN), microbial biomass (SBMC), active carbon (AC), cold (CWC) and hot (HWS) salt water extractable carbon, carbon pool index (CPI), nitrogen pool index (NPI), carbon lability index (CLI), and carbon management index (CMI) under different phases of a rainfed no-till soybean-corn rotation in Alabama (2012 to 2016).**

| Treatment | Variable response | | | | | | | | | | | | | | | | |
|---|---|---|---|---|---|---|---|---|---|---|---|---|---|---|---|---|---|
| | SOC | TN | SMBC | SMBC: | AC | CWC | HWC | CPI | NPI | CLI | | | | CMI | | | |
| | | | | SOC (%) | | | | | | SMBC | AC | CWC | HWC | SMBC | AC | CWC | HWC |
| Gypsum (Mg/ha) | (g/kg) | | (mg/kg) | | (mg/kg) | | | | | | | | | | | | |
| 0 | 6.3a$^{¥}$ | 0.53ab | 103b | 2.05ab | 182b | 10.2a | 30.2b | 1.00a | 1.00b | 1.06b | 1.01a | 1.00a | 0.96a | 1.06b | 1.01a | 1.00a | 0.96a |
| 1.1 | 6.4a | 0.50b | 119ab | 2.46a | 192ab | 12.4a | 36.2ab | 1.01a | 0.94c | 1.08ab | 1.02a | 0.99a | 0.98a | 1.09ab | 1.03a | 1.00a | 0.99a |
| 2.2 | 6.5a | 0.61a | 138a | 2.5a | 221a | 10.5a | 38.1a | 1.03a | 1.15a | 1.16a | 1.07a | 1.01a | 1.03a | 1.19a | 1.10a | 1.04a | 1.06a |
| **Crop rotation** | | | | | | | | | | | | | | | | | |
| CS | 7.7A$^{\neq}$ | 0.53A | 122A | 1.75B | 247A | 11.5A | 35.2A | 1.23A | 1.00A | 1.18A | 1.09A | 1.04A | 1.05A | 1.45A | 1.34A | 1.28A | 1.29A |
| SC | 7A | 0.55A | 111A | 1.81B | 203A | 10.2A | 31.5A | 1.08B | 1.04A | 1.23A | 1.12A | 1.08A | 1.10A | 1.33B | 1.21B | 1.17B | 1.19B |
| SS | 4.7B | 0.56A | 128A | 3.44A | 144B | 11.4A | 37.7A | 0.74C | 1.06A | 1.26A | 1.13A | 1.10A | 1.13A | 0.93C | 0.84C | 0.81C | 0.84C |
| **Cover crop** | | | | | | | | | | | | | | | | | |
| Control | 6.3x$^{£}$ | 0.50x | 118x | 2.27x | 195x | 11.1x | 35.6x | 1.00x | 0.95y | 1.29x | 1.12x | 1.16x | 1.15x | 1.29x | 1.12x | 1.16x | 1.15x |
| Rye | 6.5x | 0.60x | 123x | 2.40x | 202x | 10.9x | 34.0x | 1.03x | 1.11x | 1.28x | 1.11x | 1.17x | 1.16x | 1.32x | 1.14x | 1.21x | 1.19x |

$^{¥}$ Means separated by same lower-case letter (a, b, and c) in each column were nonsignificant (at p≤0.05) among gypsum application rates by the LSD test.

$^{\neq}$ Means separated by same upper-case letter (A, B and C) in each column were nonsignificant (at p≤0.05) among crop rotations by the LSD test.

$^{£}$ Means separated by same lower-case letter (x and y) in each column were nonsignificant (at p≤0.05) between cover crops by the LSD test.

**Table 2. Gypsum and cover crop impacts on total soil organic C (SOC), total nitrogen (TN), microbial biomass (SBMC), active carbon (AC), cold (CWC) and hot (HWC) salt water extractable carbon, carbon pool index (CPI), nitrogen pool index (NPI), carbon lability index (CLI), and carbon management index (CMI) under different phases of a rainfed no-till soybean-corn rotation in Ohio-Hoytville (2012 to 2016).**

| Treatment | Variable response | | | | | | | | | | | | | | | | |
|---|---|---|---|---|---|---|---|---|---|---|---|---|---|---|---|---|---|
| | SOC | TN | SMBC | SMBC: | AC | CWC | HWC | CPI | NPI | CLI | | | | CMI | | | |
| | | | | SOC (%) | | | | | | SMBC | AC | CWC | HWC | SMBC | AC | CWC | HWC |
| Gypsum (Mg/ha) | (g/kg) | | (mg/kg) | | (mg/kg) | | | | | | | | | | | | |
| 0 | 16.8a[¥] | 1.99a | 149b | 0.89b | 537b | 17.6ba | 49.1a | 1.01a | 1.00a | 0.97a | 1.00a | 1.05a | 1.02a | 0.98b | 1.01a | 1.06a | 1.03a |
| 1.1 | 17.6a | 2.06a | 162ab | 0.93ab | 547ab | 20.7a | 55.7ab | 1.06a | 1.03a | 1.03a | 1.00a | 1.06a | 1.05a | 1.09a | 1.06a | 1.12a | 1.11a |
| 2.2 | 17.5a | 2.04a | 188a | 1.08a | 562.4a | 19.6ab | 60.5a | 1.05a | 1.02a | 1.05a | 0.99a | 1.07a | 1.06a | 1.10a | 1.04a | 1.12a | 1.11a |
| Crop rotation | | | | | | | | | | | | | | | | | |
| CS | 17.2A[≠] | 2.05A | 171A | 0.99A | 532B | 20.1A | 57.3A | 1.03A | 1.02A | 1.09B | 1.00A | 1.09A | 1.09A | 1.12B | 1.03A | 1.12A | 1.12A |
| SC | 17.2A | 2.01A | 157A | 0.91A | 553AB | 18.9A | 52.8A | 1.03A | 1.00A | 1.17AB | 1.00A | 1.12A | 1.14A | 1.21AB | 1.03A | 1.15A | 1.17A |
| SS | 17.6A | 2.04A | 171A | 0.99A | 561A | 18.8A | 55.2A | 1.06A | 1.02A | 1.24A | 1.01A | 1.11A | 1.15A | 1.31A | 1.07A | 1.18A | 1.22A |
| Cover crop | | | | | | | | | | | | | | | | | |
| Control | 17.2x[£] | 2.04x | 148y | 0.87y | 524y | 18.5x | 58.2x | 1.04x | 1.02x | 1.41x | 1.03x | 1.14x | 1.22x | 1.47x | 1.07x | 1.19x | 1.27x |
| Rye | 17.4x | 2.03x | 185x | 1.06x | 574x | 20.0x | 52.0x | 1.04x | 1.01x | 1.44x | 1.03x | 1.12x | 1.22x | 1.50x | 1.07x | 1.16x | 1.27x |

[¥] Means separated by same lower-case letter (a, b, and c) in each column were nonsignificant (at p≤0.05) among gypsum application rates by the LSD test.

[≠] Means separated by same upper-case letter (A, B and C) in each column were nonsignificant (at p≤0.05) among crop rotations by the LSD test.

[£] Means separated by same lower-case letter (x and y) in each column were nonsignificant (at p≤0.05) between cover crops by the LSD test.

increase being greater for the 2.2 versus the 1.1 Mg/ha gypsum treatment. The size of the biological C pool (SMBC:SOC) also increased with gypsum application at all sites except Ohio-Piketon. Gypsum applied at the 2.2 kg/ha significantly increased both active C and hot-water C pools.

The CPI, (i.e., a measure of SOC accumulation or depletion) in general, did not change significantly. The NPI value, however, was increased significantly, but only in Alabama (by 15%) by the 2.2 Mg/ha gypsum application (Table 1). The SOC lability did not vary consistently

**Table 3. Gypsum and cover crop impacts on total soil organic C (SOC), total nitrogen, microbial biomass (SBMC), active carbon (AC), cold (CWC) and hot (HWC) salt water extractable carbon, carbon pool index (CPI), nitrogen pool index (NPI), carbon lability index (CLI) and carbon management index (CMI) under different phases of a rainfed no-till soybean-corn rotation in Indiana (2012 to 2016).**

| Treatment | Variable response | | | | | | | | | | | | | | | | |
|---|---|---|---|---|---|---|---|---|---|---|---|---|---|---|---|---|---|
| | SOC | TN | SMBC | SMBC: | AC | CWC | HWC | CPI | NPI | CLI | | | | CMI | | | |
| | | | | SOC (%) | | | | | | SMBC | AC | CWC | HWC | SMBC | AC | CWC | HWC |
| Gypsum (Mg/ha) | (g/kg) | | (mg/kg) | | (mg/kg) | | | | | | | | | | | | |
| 0 | 12.7a[¥] | 1.39a | 143c | 1.23c | 405a | 27.8a | 40.0b | 0.99a | 1.00a | 0.99a | 1.07a | 1.05a | 0.98a | 0.98a | 1.06a | 1.04a | 0.97a |
| 1.1 | 12.4a | 1.38a | 211b | 1.84b | 403a | 26.8a | 44.8b | 0.96a | 0.99a | 1.04a | 1.07a | 1.05a | 1.01a | 1.00a | 1.03a | 1.01a | 0.97a |
| 2.2 | 12.6a | 1.41a | 296a | 2.49a | 405a | 31.1a | 56.3a | 0.98a | 1.01a | 1.08a | 1.07a | 1.04a | 1.02a | 1.06a | 1.05a | 1.02a | 1.00a |
| Crop rotation | | | | | | | | | | | | | | | | | |
| CS | 14.4A[≠] | 1.58A | 193B | 1.42C | 455A | 28.6A | 45B | 1.12A | 1.14A | 1.08A | 1.07A | 1.02A | 1.02A | 1.21A | 1.20A | 1.14A | 1.14A |
| SC | 11.7B | 1.32B | 199B | 1.8B | 365B | 27.2A | 44.1B | 0.91B | 0.95B | 1.05A | 1.06A | 1.06A | 1.01A | 0.96B | 0.96B | 0.96A | 0.92B |
| SS | 11.6B | 1.29B | 259A | 2.33A | 393B | 29.9A | 52.0A | 0.90B | 0.92B | 1.06A | 1.06A | 1.10A | 1.03A | 0.95B | 0.95B | 0.99A | 0.93B |
| Cover crop | | | | | | | | | | | | | | | | | |
| Control | 12.5x[£] | 1.41x | 203y | 1.80x | 404x | 28x | 45.3x | 0.97x | 1.00x | 1.06x | 1.06x | 1.12x | 1.05x | 1.03x | 1.03x | 1.09x | 1.02x |
| Rye | 12.7x | 1.40x | 231x | 1.91x | 404x | 29.1x | 48.8x | 0.98x | 1.00x | 1.10x | 1.07x | 1.12x | 1.07x | 1.08x | 1.05x | 1.10x | 1.05x |

[¥] Means separated by same lower-case letter (a, b, and c) in each column were nonsignificant (at p≤0.05) among gypsum application rates by the LSD test.

[≠] Means separated by same upper-case letter (A, B and C) in each column were nonsignificant (at p≤0.05) among crop rotations by the LSD test.

[£] Means separated by same lower-case letter (x and y) in each column were nonsignificant (at p≤0.05) between cover crops by the LSD test.

**Table 4. Gypsum and cover crop impacts on total soil organic C (SOC), total nitrogen, microbial biomass (SBMC), active carbon (AC), cold (CWC) and hot (HWC) salt solution extractable carbon, carbon pool index (CPI), nitrogen pool index (NPI), carbon lability index (CLI) and carbon management index (CMI) under different phases of a rainfed no-till soybean-corn rotation in Ohio-Piketon (2012 to 2016).**

| Treatment | Variable response | | | | | | | | | | | | | | | | |
|---|---|---|---|---|---|---|---|---|---|---|---|---|---|---|---|---|---|
| | SOC | TN | SMBC | SMBC: | AC | CWC | HWC | CPI | NPI | CLI | | | | CMI | | | |
| | | | | SOC (%) | | | | | | SMBC | AC | CWC | HWC | SMBC | AC | CWC | HWC |
| Gypsum (Mg/ha) | (g/kg) | | (mg/kg) | | (mg/kg) | | | | | | | | | | | | |
| 0 | 6.9a¥ | 0.86a | 205a | 3.19ab | 338b | 15.1a | 58.7a | 1.00a | 1.00a | 1.00a | 1.02a | 0.96a | 0.89a | 1.00a | 1.02a | 0.96a | 0.89a |
| 1.1 | 6.9a | 0.85a | 201a | 3.39a | 368ab | 14.7a | 57.6a | 1.02a | 1.01a | 1.04a | 1.02a | 0.96a | 0.87a | 1.05a | 1.04a | 0.98a | 0.88a |
| 2.2 | 7a | 0.88a | 204a | 3.01b | 418a | 15.8a | 59.3a | 1.03a | 1.04a | 1.04a | 1.03a | 0.97a | 0.87a | 1.07a | 1.06a | 1.00a | 0.90a |
| **Crop rotation** | | | | | | | | | | | | | | | | | |
| CS | 7.7A≠ | 0.91A | 202A | 2.7B | 440A | 14.9A | 57.9AB | 1.11A | 1.09A | 1.04A | 1.02A | 0.96A | 0.87A | 1.16A | 1.14A | 1.07A | 0.97A |
| SC | 6.7B | 0.83A | 232A | 3.50A | 391AB | 15.7A | 65.2A | 1.00B | 0.99B | 1.04A | 1.04A | 0.97A | 0.87A | 1.04B | 1.04B | 0.97B | 0.87B |
| SS | 6.4B | 0.83A | 177B | 3.38A | 293B | 15A | 52.6B | 0.93B | 0.99B | 1.01A | 1.02A | 0.96A | 0.86A | 0.95B | 0.95B | 0.90B | 0.80B |
| **Cover crop** | | | | | | | | | | | | | | | | | |
| Control | 6.9x£ | 0.87x | 193x | 3.05y | 370x | 15.2x | 61.0x | 1.02x | 1.01x | 1.01x | 1.05x | 0.98x | 0.86x | 1.03x | 1.07x | 1.00x | 0.87x |
| Rye | 6.9x | 0.85x | 215x | 3.35x | 380x | 15.2x | 56.2x | 1.02x | 1.03x | 1.01x | 1.04x | 0.95x | 0.85x | 1.03x | 1.06x | 0.97x | 0.86x |

¥ Means separated by same lower-case letter (a, b, and c) in each column were nonsignificant (at p≤0.05) among gypsum application rates by the LSD test.

≠ Means separated by same upper-case letter (A, B and C) in each column were nonsignificant (at p≤0.05) among crop rotations by the LSD test.

£ Means separated by same lower-case letter (x and y) in each column were nonsignificant (at p≤0.05) between cover crops by the LSD test.

except for a slight improvement (by 8 to 10%), due to a change in the SMBC value in Alabama, where the 2.2 Mg gypsum/ha was applied (Table 1). In contrast, the CMI, a composite measure of SOC accumulation and lability, was significantly affected (by 12 to 13%) via a change in SMBC by the impact of the 2.2 Mg/ha gypsum application in both Alabama and Ohio-Hoytville sites.

When the data on SOC pools, total N, and SOC lability were normalized at each site, followed by an across-sites analyses (i.e., a meta-analysis), SMBC (by 41%), SMBC:SOC (by

**Table 5. Gypsum and cover crop impacts on normalized values of total soil organic C (SOC), total nitrogen (TN), microbial biomass (SBMC), active carbon (AC), cold (CWC) and hot (HWC) salt water extractable carbon, carbon pool index (CPI), nitrogen pool index (NPI), carbon lability index (CLI) and carbon management index (CMI) under different phases of a rainfed no-till soybean-corn rotation when averaged across all sites (2012 to 2016).**

| Treatment | Variable response | | | | | | | | | | | | | | | | |
|---|---|---|---|---|---|---|---|---|---|---|---|---|---|---|---|---|---|
| | SOC | TN | SMBC | SMBC: | AC | CWC | HWC | CPI | NPI | CLI | | | | CMI | | | |
| | | | | SOC (%) | | | | | | SMBC | AC | CWC | HWC | SMBC | AC | CWC | HWC |
| Gypsum (Mg/ha) | (g/kg) | | (mg/kg) | | (mg/kg) | | | | | | | | | | | | |
| 0 | 1.01a¥ | 1.00a | 0.96c | 0.96c | 1.02b | 1.02a | 0.97c | 1.00a | 1.00a | 1.01a | 1.03a | 1.02a | 0.96a | 1.00a | 0.98a | 1.00a | 1.00a |
| 1.1 | 1.02a | 0.99a | 1.13b | 1.16b | 1.06ab | 1.11a | 1.08b | 1.01a | 0.99a | 1.05a | 1.03a | 1.02a | 0.98a | 1.06a | 1.00a | 1.02a | 1.04a |
| 2.2 | 1.03a | 1.05a | 1.37c | 1.32a | 1.15a | 1.10a | 1.20a | 1.02a | 1.06a | 1.08a | 1.04a | 1.02a | 1.00a | 1.07a | 1.03a | 1.05a | 1.07a |
| **Crop rotation** | | | | | | | | | | | | | | | | | |
| CS | 1.14A≠ | 1.05A | 1.12B | 0.96C | 1.21A | 1.09A | 1.08A | 1.12A | 1.06A | 1.10A | 1.05A | 1.03A | 1.01A | 1.24A | 1.19A | 1.21A | 1.25A |
| SC | 1.00AB | 0.99A | 1.12B | 1.08B | 1.07AB | 1.05A | 1.05A | 1.01B | 0.99A | 1.12A | 1.06A | 1.06A | 1.03A | 1.03B | 1.00B | 1.04B | 1.06B |
| SS | 0.92B | 0.99A | 1.22A | 1.40A | 0.94B | 1.09A | 1.11A | 0.91B | 0.99A | 1.14A | 1.06A | 1.07A | 1.04A | 0.95B | 0.89C | 0.93C | 0.96C |
| **Cover crop** | | | | | | | | | | | | | | | | | |
| Control | 1.01x£ | 0.99x | 1.13x | 1.13x | 1.05x | 1.06x | 1.06x | 1.01x | 1.00x | 1.19x | 1.07x | 1.10x | 1.07x | 1.11x | 1.03x | 1.10x | 1.12x |
| Rye | 1.03x | 1.03x | 1.18x | 1.16x | 1.10x | 1.09x | 1.10x | 1.02x | 1.04x | 1.21x | 1.06x | 1.09x | 1.08x | 1.14x | 1.04x | 1.10x | 1.14x |

¥ Means separated by same lower-case letter (a, b, and c) in each column were nonsignificant (at p≤0.05) among gypsum application rates by the LSD test.

≠ Means separated by same upper-case letter (A, B and C) in each column were nonsignificant (at p≤0.05) among crop rotations by the LSD test.

£ Means separated by same lower-case letter (x and y) in each column were nonsignificant (at p≤0.05) between cover crops by the LSD test.

36%), active C (by 13%), and hot-water C (by 23%) showed positive responses to the 2.2 Mg/ha gypsum, but without any significant increase in SOC accumulation or lability (Table 5).

A significant increase in the SMBC, SMBC:SOC, active C, and hot-water C indicate a biochemical response to the gypsum amendments that contributes to improved soil quality. Gypsum is more soluble (by about 150 to 200 times) than lime, another Ca-containing amendment that increases the mobility of $Ca^{+2}$, $Mg^{+2}$, and $K^+$ associated with $SO_4^{-2}$ in the soil profile [19, 21]. This influences soil chemical and physical properties that are more favorable for SMBC and associated biological activities [16]. Other studies, however, have reported that soil biological and chemical properties were not consistently influenced by gypsum or similar types of soil chemical amendments [24, 37].

While the SMBC is only a small fraction (~ 5%) of SOC, it acts as a proactive biocatalyst for residue decomposition, nutrient recycling, and aggregate formation associated with soil quality [5, 38]. An increase in SMBC translates to an enlargement in the size of the biologically labile pool of SOC (SMBC:SOC), which shows a positive response with improved C-use efficiency towards SOC accumulation and lability [4, 5, 15].

Several studies have reported that the SMBC and community structures act indirectly and positively to the changing or balancing of soil geochemical and physical properties by agricultural management practices [24, 39]. Gypsum amendments combined with transitional no-till seem to affect the SMBC diversity, including fungi dominance [39]; however, as reported, not all microbes are sensitive to gypsum application [24]. Our results showed that gypsum, applied at the 2.2 Mg/ha rate, exerted complementary effects on the no-till performance to increase SMBC and the size of the biologically labile pool of SOC (SMBC:SOC). This is thought to be primarily mediated by shifting a change towards a fungi-based food web and an improvement in soil functions under no-till [16, 17, 39]. Likewise, significantly higher active C and hot-water C were observed due to synergistic effects of microbes-residue interactions, including both agronomic and cover crops' response to gypsum application. A significantly higher SMBC with improved C-use efficiency (higher SMBC:SOC) may have acted positively to affect a higher level of formation of active C and hot-water C via a change in SMBC [8, 11, 15]. This is attributed to a greater release or formation of microbial metabolites and root exudates, and production of fine roots of both agronomic and cover crops as influenced by Ca, Mg, and S nutrition from the gypsum amendments [9, 16]. As the SMBC and SMBC:SOC values were higher in the 2.2 Mg/ha gypsum rate, this indicates the prolonged and cumulative residual impact of gypsum applied as a proactive soil chemical amendment.

Significantly higher values of CMI in the marginal soils of Alabama suggests that a change is occurring in SOC lability, but not in significant accumulation. This was mostly attributed to an increase in the size of the SMBC pool and its efficient C anabolism. Soils with higher CMI values are considered to be better managed with higher C-use efficiency towards improved soil quality [16, 40]. In this study, a consistent effect of the 2.2 Mg/ha gypsum application rate on SMBC, SMBC:SOC, and CMI was observed when meta-analysis of data was performed across the sites.

## Crop rotation impacts on soil carbon pools and lability

The impact of CS rotation in different phases compared to SS variably affected the SOC pools and total N (Tables 1–5). The SOC values were higher by 49 to 60% under both CS phases than under SS in Alabama, and were also higher, to a lesser degree, under CS-S and SS rotations, respectively, in Indiana and Ohio-Piketon. Total N was also significantly higher in Indiana by more than 12% in soils under CS-C than in soils under both CS-S and SS rotations.

The SMBC values, in contrast, were significantly higher in Indiana under SS than under both phases of the CS rotation, but significantly lower in Ohio-Piketon under SS than under

both CS phases (Tables 4 and 5). A positive impact of SS compared to CS phases on SMBC: SOC was also observed at all sites except Ohio-Hoytville. Like SMBC, the active C values were similarly greater under both CS phases than under SS at all sites except Indiana. Crop rotation also variably influenced hot-water C.

The CPI was significantly higher by 11 to 49% at all sites except Ohio-Hoytville under both CS phases when compared with SS (Tables 1, 3 and 4). In contrast, the NPI was significantly higher in Indiana (by 19 to 22%) and Ohio-Piketon (by 10%) under both CS phases compared to SS. The CMI, in contrast, did vary significantly by the impact of crop rotation at all sites. The CMI, as influenced by SMBC, was higher under both CS phases compared to SS at all sites except Ohio-Hoytville, where an opposite trend was observed. The CMI values, as influenced via active C and cold and hot-water C, had similarly higher values under both CS phases than SS at all sites except Ohio-Hoytville.

When the data were normalized at each site, followed by across the sites (i.e., meta-analysis), the SOC, active C, CPI, and CMIs values increased under both CS phases when compared to SS. An opposite trend, however, was observed on SMBC and SMBC:SOC by crop rotation (Table 5). The values of CMIs via a change in SMBC was consistently higher than active C and cold and hot-water C under both CS phases when compared to SS.

In agroecosystems, the primary input of SOC is shoot and root biomass of crops. An increase in SOC across the sites, except Ohio-Hoytville, under CS-C compared to SS was due to increased deposition of C inputs that consists of larger amounts of C-enriched corn biomass followed by lesser amounts of highly decomposable N-rich soybean biomass. In comparison, there was much less total C input under SS. Crop residues with a high C:N ratio that are returned to soil are expected to decompose more slowly and have a longer residence time in marginal soils like Alabama and Ohio-Piketon [9, 25, 41]. Furthermore, the combination between residue retention and transitional no-till has a greater impact in accumulating SOC under CS-C than under SS.

A variability in SMBC and SMBC:SOC values caused by crop rotations at different sites suggests a proportionally variable shift in the biologically labile pool of SOC as influenced by the amount and diversity of residue C and nutrients returning to the soil. An increase in SMBC and SMBC:SOC across the experimental sites over time was due to surface accumulation of C-enriched residue from corn and N-enriched residue from soybeans [25, 26, 42]. Moreover, inherent regional and site variability such as climatic conditions, geochemical, soil types and texture, and moisture dynamics are also thought to have influenced the variability in SMBC and SMBC:SOC values [5, 34]. However, several studies have suggested that the effects of residue quality are mostly short-term, and that all residues become chemically similar once processed by SMBC [42]. This has been supported by the emerging new concepts of SOC formation, which suggest that the labile pools of SOC are composed mostly of microbial cells and their metabolites [9, 42].

Significant increase in active C values under both CS phases was also associated with the greater amount of corn and soybean residues returning to the soil than the smaller amount of soybean residues over time under SS. The active C values, a measure of labile SOC, is closely associated with SOC and SMBC, and an increase or a depletion in SOC and SMBC is also expected to invariably affect the active C content [8, 12, 16]. However, as previously stated, the amount and quality of residue returning, and regional and site variability, are also expected to play important roles. Given the critical impact that SMBC has in processing crop residues, and indirectly contributing their cells and their metabolites to active C and SOC formation [9, 16, 43], the positive effects of crop rotation may have led to greater SOC accumulation in Alabama and Ohio-Piketon soils than in Ohio-Hoytville and Indiana soils. This was reflected in the higher CMI values via CPI observed for the CS rotations.

The highest CMIs observed under both CS phases followed by SS across the sites were attributed to the higher CPI values that were impacted by the greater amounts of C released upon decomposition of diverse crop residues as microbial cells and metabolites, fine roots production, root exudates, and humus. Because CMI is an integrated measure for both the quantitative and qualitative nature of SOC, soils with higher CMIs, as influenced by greater availability of labile C, are considered to be better managed to improve soil quality [4, 16, 40]. The CMI may induce changes in other soil functional properties including biodiversity and efficiency, enzyme activities, and N availability in response to balance among microbes and plants, residue decomposition, and SOM formation and mineralization.

In our study, the changes observed in CMI values, as induced by agricultural management practices, were of a higher degree of CPI than changes on SOC lability (CLI), a tendency also observed in other studies [4, 16, 40]. In other words, the CPI values control the values of the CMI. Thus, calculating CPI and CMI values provide a way to detect early changes in SOC lability and accumulation or depletion, which are undetectable when considering SOC values.

### Cover crop impacts on soil carbon pools and lability

Integrating rye as a winter cover crop into the crop rotation exerted limited impacts on SOC pools, total N, and SOC lability (Tables 1–5). Rye significantly increased the SMBC (by 14 to 24%) in Ohio-Hoytville and Indiana soils, respectively, when compared to their control soils. The SMBC:SOC was also significantly increased by the impact of rye in Ohio-Hoytville and Ohio-Piketon. Likewise, the active C was increased by the rye at all sites, but only significantly at Ohio-Hoytville (Table 2). While the CPI did not change, the NPI values were increased (by 16%) under rye, but only in Alabama (Table 1). Normalization of data among and across the sites showed that rye promoted only a slight increase in SOC pools, total N, and SOC lability, and this increase was not consistently significant (Table 5).

A non-significant increase in SOC pools, TN, and SOC lability was reported by the impact of rye across all sites [29, 43]. This contrasts with expected results that when long-term rotations include a cover crop, there will be pronounced effects leading to an influence on SOC pools. However, we observed limited impacts of rye when used as a winter cover crop due to moderate growth and the high C:N values of cover crop biomass inputs. While rye is a preferred species of winter cover crop across the Midwest, its biomass contribution is highly constrained by the limited growing degree days during the winter and early spring months before planting of corn and soybean crops [28, 29, 43]. We believe that a moderate amount of high C:N ratio (> 80:1) biomass contribution from rye at our sites, when compared to the larger amount of background agronomic crop residues (such as corn), did not contribute to significantly influence SMBC, SOC pools, and lability. This suggests that to take advantage of management practices that include cover crops, the cover crops grown must be adapted to the soil types and climatic zones where they are grown, so that a substantial amount of cover crop biomass is produced. This may be done by using cover crop mixtures and by other practices that promote early and more significant cover crop growth [29].

### Interaction of gypsum and cover crops on soil carbon pools and lability

In general, there were few significant interactive effects noted due to treatment combinations. One exception to this was gypsum by cover crop interaction. This was found at most individual sites, but not across all sites or via the meta-analysis. At the individual sites, it seems like the rotation effect is more pronounced at the highest gypsum application rate of 2.2 Mg/ha when compared to the lower gypsum rate or no gypsum application.

### Relationship between soil carbon and nitrogen pool (accumulation) indices

When plotted, the SOC and TN pools at each site and across the sites showed a positive and significant linear relationship between them (Fig 2). The exception to this observation was at the Alabama site. The SOC significantly accounted for 88, 89, and 90% of the TN variability in Ohio-Hoytville, Indiana, and Ohio-Piketon, respectively, but at the Alabama site, the $R^2$ value that related SOC and TN was only 0.24. A meta-analysis of data from all sites showed that the SOC significantly explained 89% of the variability in TN with a C:N slope of 10:1.1, which closely conforms to a C:N ration of 10:1 that is accepted for SOM [10, 44].

A similar response between the CPI and NPI values was observed at each site and among the sites (Fig 3). The CPI, a measure of SOC accumulation or depletion over the control, significantly accounted for 77 to 82% of the variability in the NPI, a measure of TN accumulation or depletion over the control, and/or vice-versa in Indiana, Ohio-Piketon, and Ohio-Hoytville. The CPI non-significantly accounted for only 24% of the NPI variability in Alabama. When pooling all the normalized data, 90% of the variability between the CPI and NPI values was accounted for via a strong non-linear relationship. The slope for this relationship had a ratio of 10:0.81, which is slightly different than the standard C:N slope (10:1) in SOM.

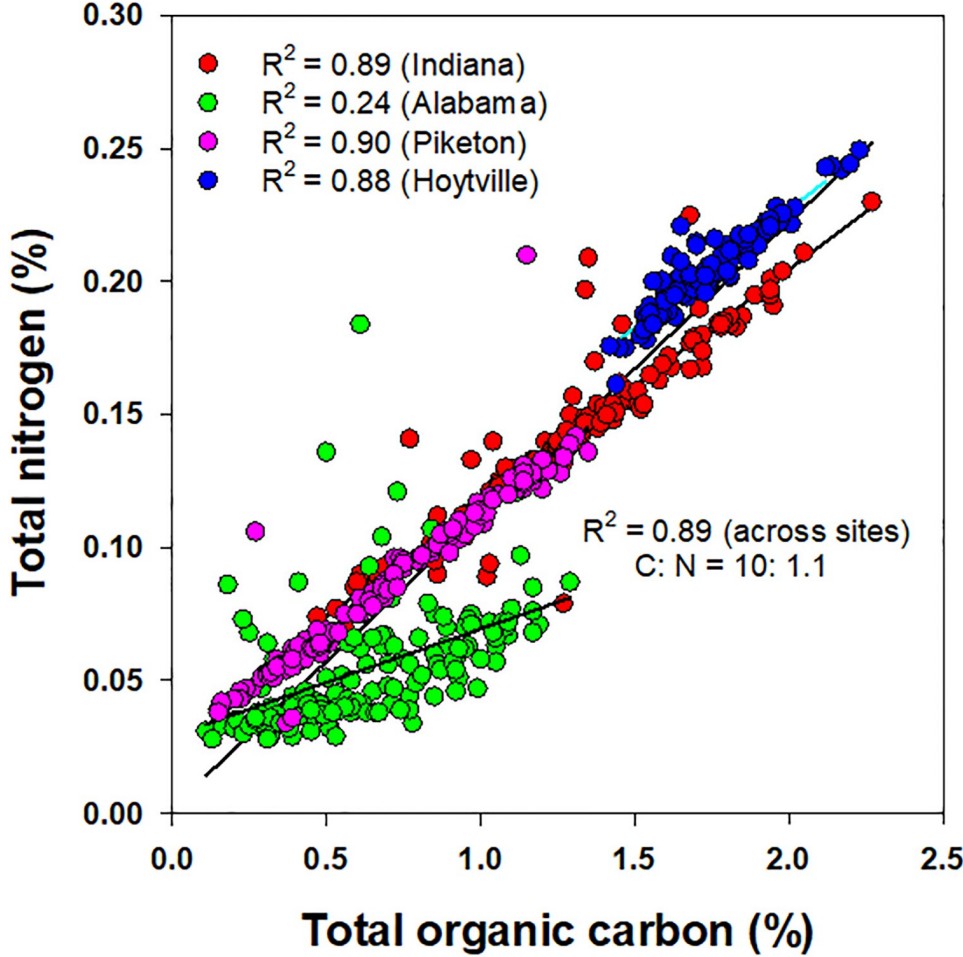

**Fig 2. Relationship between soil total organic carbon (SOC) and total nitrogen (TN) content for a gypsum-amended rainfed transitional no-till soybean-corn rotation with cover crop in Alabama (Shorter), Indiana (Farmland), and Ohio (Hoytville and Piketon) and averaged across all data.**

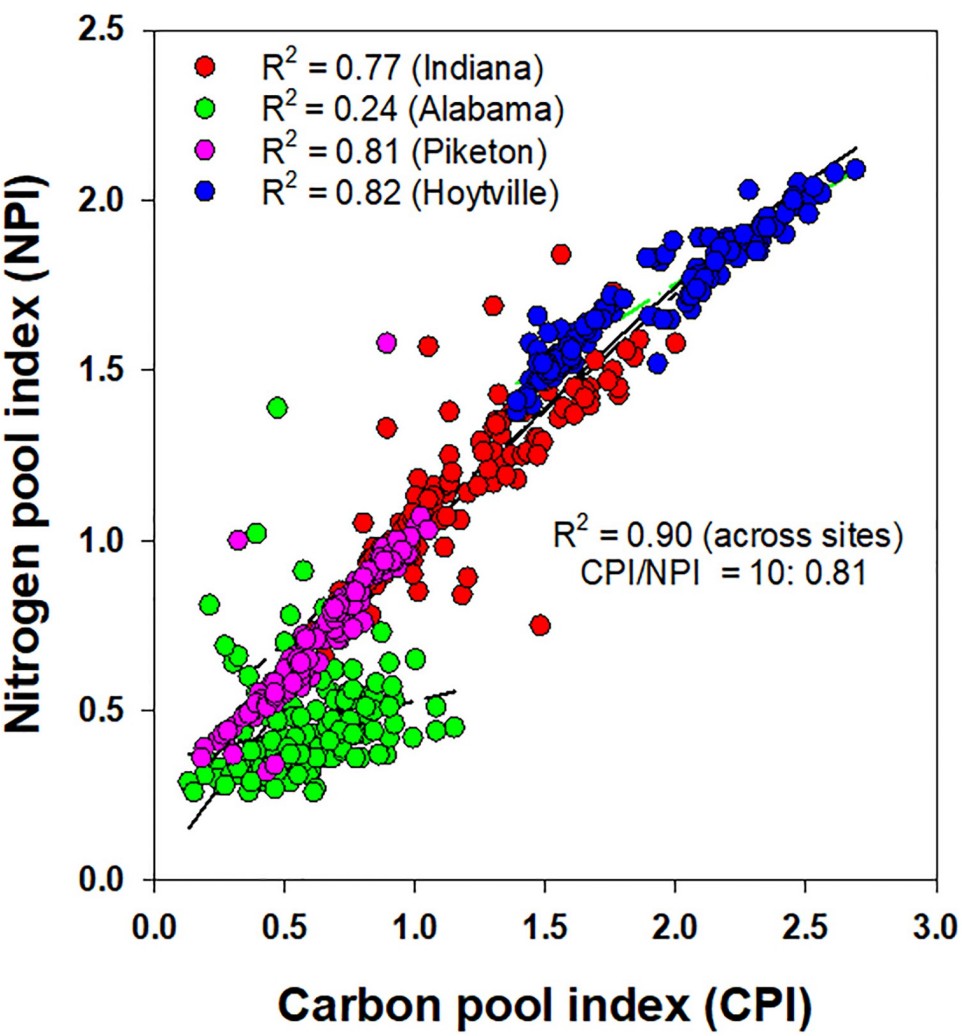

**Fig 3. Relationship between soil carbon and nitrogen pool indices for a gypsum-amended rainfed transitional no-till soybean-corn rotation with cover crop in Alabama (Shorter), Indiana (Farmland), and Ohio (Hoytville and Piketon) and averaged across all data.**

A significant relationship observed between SOC and TN confirms that the C and N remain strongly coupled in SOM at all sites except marginal soils in Alabama. Although it is reported that C and N remained coupled in SOM, the ratios of C and N may be significantly changed in C-enriched surface soils. The calculated C:N ratios derived from the best-fitted, non-linear relationship between the CPI and NPI values suggest a slightly greater C enrichment in SOM when compared to the N accumulation. This is attributed to a greater amount of surface accumulation of C-enriched unfragmented crop residues (such as CS phases) including rye biomass when used as a cover crop under transitional no-till across the sites. A greater variability observed in the NPI values when correlated with the CPI was related to the more dynamic and labile nature of the total N compared with the SOC. This may also partially account for slightly wider CPI:NPI ratios compared to the conventional C:N ratios observed across the sites.

In accordance with biochemical stoichiometry, the formation of SOM requires a certain amount of N and other nutrients in a fixed ratio with SOC [45]. An increased flow of C-enriched residues (such as corn) in both CS phases than in SS is expected to reduce

decomposition of residues and restrict N availability, which may potentially lead to a progressive N limitation in soil [46, 47]. Without new or more N input, the formation of SOM and availability of N is expected to decrease over time under climate change effects. A higher C:N ratio SOM has been observed under $CO_2$ enrichment experiments, which result in slow N mineralization, thus, affecting the release N from SOM to support plant uptake [48]. A higher CPI:NPI observed in our study suggests that a transitional no-till soil ecosystem under climate change effects would need more balanced N fertilization to couple with SOC dynamics for improved soil quality functions.

## Relationship between soil carbon accumulation and lability

The CPI, as a measure of SOC accumulation or depletion, showed a significant non-linear inverse relationship with the SOC lability, except for active C (Fig 4b), within and across the sites (Fig 4). The CPI increased with a significant non-linear decrease in SOC lability via SMB and accounted for 41% of its variability across the sites (Fig 4a). The CPI, when correlated with SOC lability associated with cold and hot-water C, accounted for a significant non-linear decrease (by 47% and 42%) of SOC lability (Fig 4c and 4d). In other words, when OC accumulates in soil, it is because its lability has decreased, presumably because the SMBC has processed all of the readily available C into a less labile form.

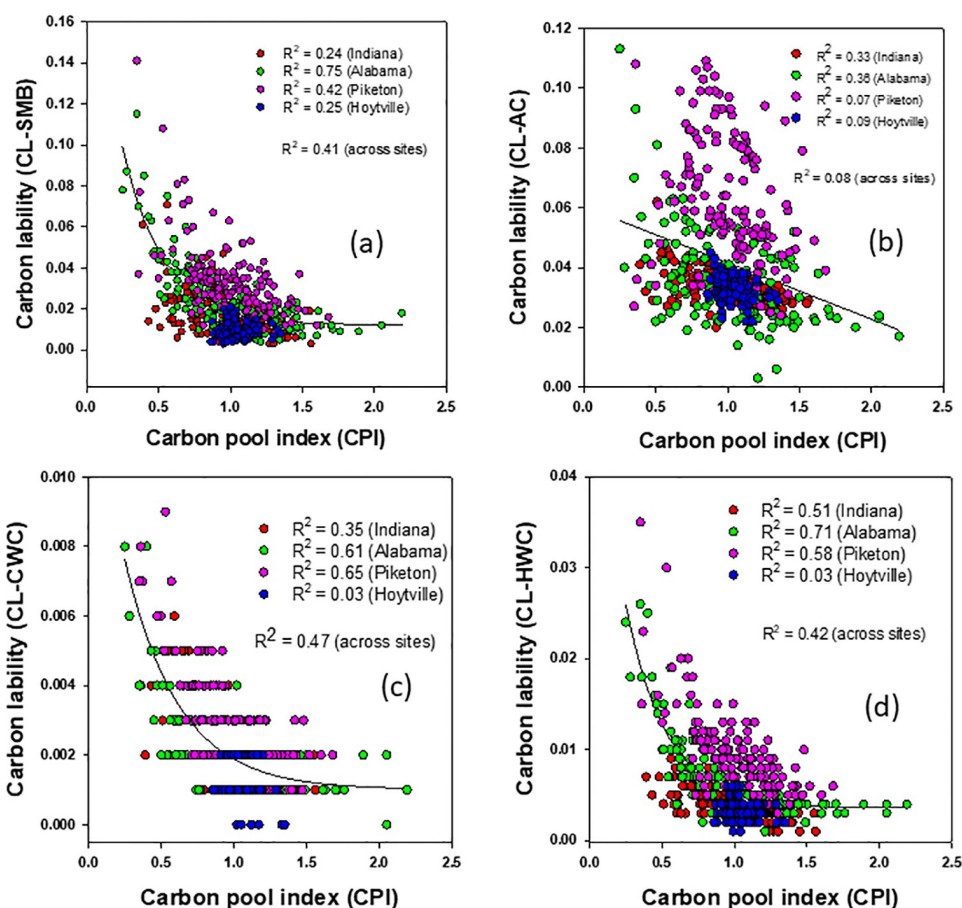

**Fig 4. Relationship between soil carbon pool index and carbon lability indices for a gypsum-amended rainfed transitional no-till soybean-corn rotation with cover crop in Alabama (Shorter), Indiana (Farmland), and Ohio (Hoytville and Piketon) and averaged across all data.**

Higher CPI values were expected to translate into higher SOC accumulation. Moreover, higher values of CPI rather than SOC lability significantly contributed to the higher CMI values with a slightly more non-labile nature of SOC. The high CPI values under both CS phases resulted from surface deposition of a greater amount of C-enriched corn and rye biomass residues with a higher lignin content (~18%), and its consequent effect on the residue and native SOM decomposition [40, 49]. Several studies have reported that surface-deposited organic residues with high C and lignin contents slowed down decomposition and, consequently, accumulated SOC with a higher proportion of non-labile C [3, 50]. Moreover, a longer retention of surface-accumulated crop residues is expected to have greater exposures to environmental variables, especially sunlight, ozone, and UV-radiation, which may lead to an alteration in residue quality and/or palatability for SBMC [51] and thus, slow down residue decomposition with a higher proportion of non-labile to labile C accumulation in SOC [3, 16].

As reported previously [32, 33], our five-year study found no supporting evidence of significantly increased crop yield or farm profitability of either gypsum application or rye cover crop. Given results reported in this manuscript, it is possible that the crop yield and economic performance of the studied practices might improve over a longer timeframe of usage.

Even though our study did not find a direct economic motive for farmer adoption of these practices, the results suggest that there may be positive economic impacts for society more broadly. A reported analysis of soil nutrient content and greenhouse gas emissions for our experimental sites found that cover crops and gypsum usage could result in at least small increases in nutrient availability and decreases in greenhouse gas emissions in certain soil and weather conditions [21, 52]. Increased carbon sequestration offers a value to society broadly through climate change remediation. Hence, policy makers may seek to encourage agricultural management practices that provide such ecosystem services. In addition, carbon markets are emerging that allow high carbon-emitting firms to purchase carbon credits from firms that can demonstrate carbon-reducing production methods or sequestration. Agricultural management practices, such as continuous no-till, crop rotation, cover crops, improved fertility practices, and use of appropriate soil chemical amendments such as gypsum may prove to have sufficient potential to increase SOC sequestration or lessen greenhouse gas emissions to the point where farmers may qualify to participate in carbon offset markets, thereby creating new sources of job and revenue.

## Conclusions

Gypsum, when applied at 2.2 Mg gypsum/ha rate, increased labile SOC pools such as SMBC, SMBC:SOC, and active C within and across the sites; however, the CPI and CMI values were not influenced consistently by gypsum application. Crop rotation positively affected SMBC, active C, and SOC lability across all the sites. In contrast, the integration of rye as a winter cover crop exerted a limited impact on the SOC pools. Overall, the SMBC, SMBC:SOC, active C, and CMI via SMBC did show a management-induced impact, thus providing an early indication of any changes in accumulation or depletion in SOC. However, SOC accumulation significantly decreased lability. The interaction of gypsum and cover crops indicates that gypsum applied at the 2.2 Mg/ha rate, coupled with CS rotation, synergistically improves soil biology and SOC dynamics under a rainfed transitional NT system.

## Supporting information

**S1 File.**
(PDF)

**S1 Table. Interactive effects of gypsum, crop rotation, and cover crop on total soil organic carbon (SOC), total nitrogen (TN), microbial biomass (SBM), metabolic quotient (qR), active carbon (AC), cold (CWC) and hot (HWC) salt water extractable carbon, carbon pool index (CPI), nitrogen pool index (NPI), carbon lability index (CLI) and carbon management index (CMI) at different soil depths under a rainfed transitioning no-till soybean-corn rotation at Alabama site (2012 to 2016).**
(DOCX)

**S2 Table. Interactive effects of gypsum, crop rotation, and cover crop on total soil organic C (SOC), total nitrogen (TN), microbial biomass (SBM), metabolic quotient (qR), active carbon (AC), cold (CWC) and hot (HWC) salt water extractable carbon, carbon pool index (CPI), nitrogen pool index (NPI), carbon lability index (CLI) and carbon management index (CMI) under a rainfed transitioning no-till soybean-corn rotation at Hoytville site (2012 to 2016).**
(DOCX)

**S3 Table. Interactive effects of gypsum, crop rotation, and cover crop on total soil organic C (SOC), total nitrogen (TN), microbial biomass (SBM), metabolic quotient (qR), active carbon (AC), cold (CWC) and hot (HWC) salt water extractable carbon, carbon pool index (CPI), nitrogen pool index (NPI), carbon lability index (CLI) and carbon management index (CMI) under a rainfed transitioning no-till soybean-corn rotation at Indiana site (2012 to 2016).**
(DOCX)

**S4 Table. Interactive effects of gypsum, crop rotation, and cover crop on total soil organic C (SOC), total nitrogen (TN), microbial biomass (SBM), metabolic quotient (qR), active carbon (AC), cold (CWC) and hot (HWC) salt water extractable carbon, carbon pool index (CPI), nitrogen pool index (NPI), carbon lability index (CLI) and carbon management index (CMI) under a rainfed transitioning no-till soybean-corn rotation at Piketon site (2012 to 2016).**
(DOCX)

**S5 Table. Interactive effects of gypsum, crop rotation, and cover crop on normalized values of on total soil organic C (SOC), total nitrogen (TN), microbial biomass (SBM), metabolic quotient (qR), active carbon (AC), cold (CWC) and hot (HWC) salt water extractable carbon, carbon pool index (CPI), nitrogen pool index (NPI), carbon lability index (CLI) and carbon management index (CMI) under a rainfed transitioning no-till soybean-corn rotation, averaged across sites (2012 to 2016).**
(DOCX)

## Author Contributions

**Conceptualization:** Khandakar R. Islam, Warren A. Dick, Dexter B. Watts, Javier M. Gonzalez, Norman R. Fausey, Randall C. Reeder, Tara T. VanToai.

**Data curation:** Dexter B. Watts, Marvin T. Batte.

**Formal analysis:** Dexter B. Watts, Javier M. Gonzalez, Dennis C. Flanagan, Tara T. VanToai, Marvin T. Batte.

**Funding acquisition:** Khandakar R. Islam, Warren A. Dick, Dexter B. Watts, Norman R. Fausey, Randall C. Reeder, Tara T. VanToai.

**Investigation:** Khandakar R. Islam, Warren A. Dick, Javier M. Gonzalez, Norman R. Fausey, Dennis C. Flanagan, Randall C. Reeder, Tara T. VanToai.

**Methodology:** Warren A. Dick, Javier M. Gonzalez, Dennis C. Flanagan, Randall C. Reeder.

**Project administration:** Khandakar R. Islam, Warren A. Dick, Norman R. Fausey, Tara T. VanToai.

**Resources:** Dexter B. Watts, Javier M. Gonzalez, Norman R. Fausey, Dennis C. Flanagan, Randall C. Reeder, Marvin T. Batte.

**Supervision:** Khandakar R. Islam, Norman R. Fausey.

**Validation:** Marvin T. Batte.

**Writing – original draft:** Khandakar R. Islam.

**Writing – review & editing:** Warren A. Dick, Dexter B. Watts, Javier M. Gonzalez, Norman R. Fausey, Dennis C. Flanagan, Randall C. Reeder, Tara T. VanToai, Marvin T. Batte.

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
