## [Decision Letter · Decision Letter 0]

1 May 2022

PONE-D-22-07617Gypsum, crop rotation and cover crop impacts on soil organic carbon and biological dynamics in rainfed transitional no-till corn-soybean systemsPLOS ONE

Dear Dr. Islam,

Thank you for submitting your manuscript to PLOS ONE. After careful consideration, we feel that it has merit but does not fully meet PLOS ONE’s publication criteria as it currently stands. Therefore, we invite you to submit a revised version of the manuscript that addresses the points raised during the review process.

We look forward to receiving your revised manuscript.

Kind regards,

Debarati Bhaduri, Ph.D.

Academic Editor

PLOS ONE

Journal Requirements:

5. Thank you for stating in your Funding Statement: 

(This study was conducted by the financial support provided by the United Soybean Board, The Ohio State University, and Ohio, Alabama, and Indiana USDA-ARS, respectively (Project # 1520-732-7226: Sustainable Production System to Improve Soybean Profitability and Soil Quality). Thanks to Emily Weaks, Hasni Jahan, Yogi Raut, and David Kost at The Ohio State University and Brenda Hofmann at USDA-ARS/Purdue University for their help to collect, process, and analyse soil samples.)

Additional Editor Comments:

Besides reviewer's comments, I would emphasize authors should revise manuscript in the following points:

1. Abstract should be written in a simplified language

2. lat-long information should be provided for all study sites, and one US maps pointing all the locations may be added for better understanding.

3. Why Table 1 is pasted in text? In fact all tables should be uploaded separately as per guidelines.

4. As it is multilocational trial, some good photographs of each location's cropping phase, and photographs taken during soil sampling can add values to the MS.

5. Number of references is too high, you can delete some multiple reference; Add some references of 2022 as well.

6. Conclusion should be little generalized (broad implications) apart from summarizing your result; Try to present in bulleted points for more clarity.

7. Please do not put the whole figure caption at P. 24, 25. Only mention Fig. 1 and Fig. 2 suitably.

8. Some track-changed words (at author's end) needs attention during revision.

Reviewers' comments:

Reviewer's Responses to Questions

**Comments to the Author**

1. Is the manuscript technically sound, and do the data support the conclusions?

Reviewer #1: No

2. Has the statistical analysis been performed appropriately and rigorously? 

Reviewer #1: No

3. Have the authors made all data underlying the findings in their manuscript fully available?

Reviewer #1: No

4. Is the manuscript presented in an intelligible fashion and written in standard English?

Reviewer #1: No

5. Review Comments to the Author

Reviewer #1: The manuscript “Gypsum, crop rotation and cover crop impacts on soil organic carbon and biological dynamics in rainfed transitional no-till corn-soybean systems” has been reviewed. The MS provides information on SOC and biological properties under different gypsum rate, crop rotation and cover crop.

After going through, the MS is having many fundamental issues which are appended below:

(i) Line 47-49: “SOC is a mixture of diverse and progressively decomposing organic compounds with different complexities and thermodynamic stabilities that greatly affects soil and environmental functions [1-4]”. This sentence is akward. The ‘mixture’ is not best fit.

(ii) Line 49: “It is a composite core indicator” needs modification. It does not sound good.

(iii) Introduction section is large. It needs to concise and simplification (for instance line 129-135 is not needed in Introduction). Also, many sentences are very complex that reduces readability.

(iv) I don’t get the point in MS of adding gypsum in the studied soil. In general, gypsum is added in sodic degraded soil for reclamation. Author must highlight the background for soil status and suggested management. It is also important to clarify whether studied region is having any sodicity problem or not. Only, for carbon sequestration gypsum is added, this statement is not correct. The crop preference and soil geochemical properties drive the management practices in a cropping ecology.

(iv) Line 177-178: Authors mentioned that ‘days under shade at room temperature, processed, and analyzed for chemical and physical analysis (Table 1). However, Table 1 is related to the ‘List of abbreviations and definitions’. The title of table of table needs modification as per text (or vice versa).

(v) Table 1. List of abbreviations and definitions: It is not a format of table (without any column and row): modify accordingly.

(vi) The major fundamental issue is the MS devoid of soil physico-chemical and biological properties of studied soil under ‘Site descriptions’ in MM section. The initial soil properties of different sites are vital for a better idea on studied soil.

(vii) I understand authors must have the yield data of crops. The yield data could be correlated with CPI, NPI for a better clarity on impact of soil properties on yield sustainability.

(vii) Excessive use of abbreviations in MS reduces the readability and clarity in MS.

(viii) In conclusion, it is mentioned that “Gypsum, crop rotation, and cover crop variably affected the depth distribution of SOC pools, TN, 607 and SOC lability without any consistent interactions within- and across the sites”. However, no table is having any mention on depth on which data presented. Only, in MM section it is mentioned 0-15 and 15-30 cm soil sampling depth (line 173). Then, where is depth wise data and how this inference was drawn.

6. PLOS authors have the option to publish the peer review history of their article (what does this mean?). If published, this will include your full peer review and any attached files.

Reviewer #1: No

---

## [Author Response · Author response to Decision Letter 0]

1 Jul 2022

Thank you so very much for your proactive comments and suggestions. Please find enclosed a revised copy of our research manuscript entitled “Gypsum, crop rotation and cover crop impacts on soil organic carbon and biological dynamics in rainfed transitional no-till corn-soybean systems” written by Islam, K.R., Dick, W.A., Watts, D.B., Gonzalez, J.M., Fausey, N.R., Flanagan, D.C., Reeder, R.C., VanToai, T., Batte, M.T. for you to evaluate its suitability of publication in PLOS ONE. 

This article is one of the outputs of the United Soybean Board funded project # 1520-732-7226; https://urldefense.com/v3/__https://www.unitedsoybean.org/__;!!KGKeukY!28VE9hc_IMjAzCa-S8fUrQhZysThVkv1FBXABtrpCwejhSI5GRcyThExS8QGh8sm9pRPnoNfwNwvx6mRJg$. However, the funder had no role in study design, data collection and analysis, decision to publish, or preparation of the manuscript.

As per your suggestions/comments, the manuscript has been revised to follow PLOS ONE's style and other requirements. Experimental data and details of the statistical analyses were presented as supporting information (S1 File and S1 – S5 Tables). 

The abstract of the manuscript was written in a simplified English language expression with a substantial deletion of abbreviations and/or acronyms. All the experimental sites including description, latitude, and longitudes were presented and explained in the map of the United States (Fig 1) and under site description. We have added cover crops, cropping phases, planting, and soil sampling photographs in the U.S. map (Fig 1). Table 1 containing abbreviations and acronyms was removed to avoid confusion.

The number of references cited was reduced to 52 (down from 79) and a few recent references from 2022 were added, including our own works. As per suggestion, the conclusion was revised to be more concise and generalized for easier understanding. A thorough revision of the manuscript was performed with especial attention to track-changed words.

If you have any questions concerning the manuscript, please feel free to communicate with me at your convenience.

Response to Reviewers' comments:

1. Is the manuscript technically sound, and do the data support the conclusions?

The manuscript must describe a technically sound piece of scientific research with data that supports the conclusions. Experiments must have been conducted rigorously, with appropriate controls, replication, and sample sizes. The conclusions must be drawn appropriately based on the data presented. Reviewer #1: No__________________________________

2. Has the statistical analysis been performed appropriately and rigorously? Reviewer #1: No

3. Have the authors made all data underlying the findings in their manuscript fully available?

The PLOS Data policy requires authors to make all data underlying the findings described in their manuscript fully available without restriction, with rare exception (please refer to the Data Availability Statement in the manuscript PDF file). The data should be provided as part of the manuscript or its supporting information, or deposited to a public repository. For example, in addition to summary statistics, the data points behind means, medians and variance measures should be available. If there are restrictions on publicly sharing data—e.g. participant privacy or use of data from a third party—those must be specified. Reviewer #1: No

4. Is the manuscript presented in an intelligible fashion and written in standard English?

PLOS ONE does not copyedit accepted manuscripts, so the language in submitted articles must be clear, correct, and unambiguous. Any typographical or grammatical errors should be corrected at revision, so please note any specific errors here. Reviewer #1: No

Response: 

A per reviewers’ comments/suggestions, the manuscript was thoroughly revised for better flow, understanding, interpretation, and conclusions presented in simplified English language. All the raw data and their in-depth statistical analysis using SAS and treatment interactions were presented with supporting information (S1 File and S1 Table – S5 Table). Anybody will be able see and use the data, if necessary (with permission). The manuscript was revised and edited by several authors and a professional editor (Bradford Sherman, The Ohio State University Communications).

Response to Specific comments

Reviewer #1: The manuscript “Gypsum, crop rotation and cover crop impacts on soil organic carbon and biological dynamics in rainfed transitional no-till corn-soybean systems” has been reviewed. The MS provides information on SOC and biological properties under different gypsum rate, crop rotation and cover crop. After going through, the MS is having many fundamental issues which are appended below:

(i) Line 47-49: “SOC is a mixture of diverse and progressively decomposing organic compounds with different complexities and thermodynamic stabilities that greatly affects soil and environmental functions [1-4]”. This sentence is akward. The ‘mixture’ is not best fit.

Response: The manuscript was thoroughly revised as per suggestion. The sentence in lines 47-49 was simplified to avoid confusion and for easier understanding.

(ii) Line 49: “It is a composite core indicator” needs modification. It does not sound good.

Response: Simplified as suggested “It is one of the core indicators……….”

(iii) Introduction section is large. It needs to concise and simplification (for instance line 129-135 is not needed in Introduction). Also, many sentences are very complex that reduces readability.

Response: Totally agreed. The introduction has been shortened. The long and complex sentences were simplified to improve flow and readability. As suggested, lines 125 through 135 have been removed.

(iv) I don’t get the point in MS of adding gypsum in the studied soil. In general, gypsum is added in sodic degraded soil for reclamation. Author must highlight the background for soil status and suggested management. It is also important to clarify whether studied region is having any sodicity problem or not. Only, for carbon sequestration gypsum is added, this statement is not correct. The crop preference and soil geochemical properties drive the management practices in a cropping ecology.

Response: Thank you for the comment. For our United Soybean Board-funded project (2012 to 2016), gypsum (flue gas desulfurized gypsum) was applied to improve marginal soil quality to support economic crop productivity, especially corn and soybeans. The soils we have selected to conduct our long-term (still on-going) experiment at all sites were marginal soils (low fertility, compacted, etc.) except Hoytville-Ohio. None of our selected sites have experienced any sodicity or salinity problems. Details of the cropping diversity, cover crops, and soils information at all sites were described in our earlier published papers (Ref. 21, 32, 33, 52). Again, gypsum was not intended to apply for sequestering carbon, but to improve marginal soil quality. A sequence of papers has been published, and more has been submitted/will be submitted for publication with information on soil fertility and chemistry including heavy metals, soil compaction and aggregate stability, greenhouse gas emissions, and inductive and deductive soil quality. 

Gypsum application has gained increasing use by farmers around the world, especially in relationship to remediating/reclaiming sodic and degraded soils. While soil carbon sequestration is a topic that has gained much attention in helping alleviate climate change, the information in the manuscript focuses on how gypsum, an agricultural soil amendment, impacts soil carbon fractions and is much needed. 

(iv) Line 177-178: Authors mentioned that ‘days under shade at room temperature, processed, and analyzed for chemical and physical analysis (Table 1). However, Table 1 is related to the ‘List of abbreviations and definitions’. The title of table of table needs modification as per text (or vice versa).

Response: Table 1 was removed to avoid confusion of widespread use of abbreviations and/or acronyms. As suggested, a large number of abbreviations were removed and explained accordingly for better flow and readability. 

(v) Table 1. List of abbreviations and definitions: It is not a format of table (without any column and row): modify accordingly.

Response: Table 1 has been removed.

(vi) The major fundamental issue is the MS devoid of soil physico-chemical and biological properties of studied soil under ‘Site descriptions’ in MM section. The initial soil properties of different sites are vital for a better idea on studied soil.

Response: Data on selected initial (2012) soil biological, chemical, and physical properties at 0-15 and 15-30 cm depths were briefly described under site description. The information was also listed in our earlier published papers (Ref. 21).

(vii) I understand authors must have the yield data of crops. The yield data could be correlated with CPI, NPI for a better clarity on impact of soil properties on yield sustainability.

Response: Thank you for the comment. Information associated with gypsum, crop rotation, and cover crop impact on crop yields was published in our earlier papers (Ref. 32 and 33). The relationship of CPI and NPI with crop yield will be presented in our upcoming “Inductive and deductive soil quality” papers (manuscript under preparation). 

(vii) Excessive use of abbreviations in MS reduces the readability and clarity in MS.

Response: As suggested, the use of abbreviations has been minimized. 

(viii) In conclusion, it is mentioned that “Gypsum, crop rotation, and cover crop variably affected the depth distribution of SOC pools, TN, 607 and SOC lability without any consistent interactions within- and across the sites”. However, no table is having any mention on depth on which data presented. Only, in MM section it is mentioned 0-15 and 15-30 cm soil sampling depth (line 173). Then, where is depth wise data and how this inference was drawn.

Response: We did point out that under no-till there will be a stratification effect on soil properties in response to lack of plowing, which was an expected outcome. That is why we did not present the soil depth information in the main tables. However, detailed information on statistical analysis associated with all the main treatments and their interactions, including soil depth, was presented in support information (S1 File and S1 Table – S5 Table).

---

## [Decision Letter · Decision Letter 1]

1 Aug 2022

PONE-D-22-07617R1Gypsum, crop rotation and cover crop impacts on soil organic carbon and biological dynamics in rainfed transitional no-till corn-soybean systemsPLOS ONE

Dear Dr. Islam,

Thank you for submitting your manuscript to PLOS ONE. After careful consideration, we feel that it has merit but does not fully meet PLOS ONE’s publication criteria as it currently stands. Therefore, we invite you to submit a revised version of the manuscript that addresses the points raised during the review process.

We look forward to receiving your revised manuscript.

Kind regards,

Debarati Bhaduri, Ph.D.

Academic Editor

PLOS ONE

Journal Requirements:

Additional Editor Comments:

Reviewer 2 has critically seen the manuscript and advised some changes that are important, and I am suggesting the authors to perform the same for further decision to be taken on the paper.

Apart from that please check again the whole paper to be technically sound, and free from from any minor errors.

Reviewers' comments:

Reviewer's Responses to Questions

**Comments to the Author**

1. If the authors have adequately addressed your comments raised in a previous round of review and you feel that this manuscript is now acceptable for publication, you may indicate that here to bypass the “Comments to the Author” section, enter your conflict of interest statement in the “Confidential to Editor” section, and submit your "Accept" recommendation.

Reviewer #1: All comments have been addressed

Reviewer #2: (No Response)

2. Is the manuscript technically sound, and do the data support the conclusions?

Reviewer #1: Yes

Reviewer #2: Partly

3. Has the statistical analysis been performed appropriately and rigorously? 

Reviewer #1: Yes

Reviewer #2: Yes

4. Have the authors made all data underlying the findings in their manuscript fully available?

Reviewer #1: Yes

Reviewer #2: Yes

5. Is the manuscript presented in an intelligible fashion and written in standard English?

Reviewer #1: Yes

Reviewer #2: Yes

6. Review Comments to the Author

Reviewer #1: (No Response)

Reviewer #2: Review report on “Gypsum, crop rotation and cover crop impacts on soil organic carbon and biological dynamics in rainfed transitional no-till corn-soybean systems” PONE-D-22-07617R1

This article is a well written manuscript based on an extensive study including field experiments at four different locations. The statistical analyses performed in the study also appears sufficient. However, I could find some issues which the authors should address before considering the article for publication.

• L 31: Write 0-15 and 15-30 cm instead of 0-15 vs. 15-30 cm

• L 46: Nitrogen pool index is mentioned in the keywords, but there is no mention of it in the abstract. At least one sentence in the abstract should mention NPI.

• The authors have responded to a query of an earlier reviewer that ‘gypsum was applied to improve marginal soil quality to support economic crop productivity’. None of the soils in this study was sodic. Then gypsum would only have supplied Ca and SO42-. First of all, if these two were not deficient in the soil (whether sufficient or deficient in the soil was not mentioned in the article), what was the original intent to apply the gypsum. Also, I could not understand how gypsum could improve different labile C pools in soil by only supplying Ca and SO42- if they were not deficient to limit microbial activity. Observing the pH, I don’t think Ca would be deficient in these soils. S may be deficient I don’t know as there is no data on that. If gypsum is not amending any inherent problem like sodicity (which is not the case here as no soil was sodic) or supplying any nutrient which is deficient, then how it has a significant effect on improving soil quality and support economic crop productivity?

• L 88-90: Mention the type of soil where such results were found

• L 99: Rewrite the sentence to avoid repetition of similar words

• L 189: Write ‘physical properties’ instead of ‘physical analysis’

• L 196: Put the starting parenthesis ‘(‘ before ‘SMBC:SOC’

• L 217-218: Authors have mentioned that “The labile C pool was considered as a portion of the SOC that was comprised of the combined pools of active C, SMBC, and cold and hot-water C.” However, active C, SMBC, and cold- and hot- water C were not extracted sequentially from the same sample; they were extracted from separate samples with different methods. Each of them may a part of the labile C; but they most likely are overlapping themselves. As they are not mutually exclusive, adding them up clearly overestimates the CL, and at the same time underestimates the non-labile C as the later was derived by subtracting CL from total SOC. Hence, the parameters derived from CL are also questionable. The authors should find a way to rectify this issue or give an appropriate rebuttal.

• Table 1, last row: This value of SMBC (1228 mg kg-1) seems too high, might be a typo, please check; If not a typo, then how come so huge difference is not significant?

• L 280-281: 2.2 kg or Mg ha-1?

• L 317-318: The authors have seen some benefits of 2.2 Mg/ha gypsum application on SMBC, SMBC:SOC, and CMI; but, without the mention of its effect on crop yield, the information remains incomplete. They should at least include the average yield of four years under different treatments.

7. PLOS authors have the option to publish the peer review history of their article (what does this mean?). If published, this will include your full peer review and any attached files.

Reviewer #1: No

Reviewer #2: No

---

## [Author Response · Author response to Decision Letter 1]

3 Sep 2022

Response to Review Comments

Reviewer # 1: (No Response)

Reviewer # 2: This article is a well written manuscript based on an extensive study including field experiments at four different locations. The statistical analyses performed in the study also appears sufficient. However, I could find some issues which the authors should address before considering the article for publication.

Response: The author(s) acknowledged reviewer’s proactive comments and/or contribution to improve the quality of the manuscript. Same credits for the academic editor.

Comment: L 31: Write 0-15 and 15-30 cm instead of 0-15 vs. 15-30 cm

Response: Thanks. Corrected accordingly.

Comment: L 46: Nitrogen pool index is mentioned in the keywords, but there is no mention of it in the abstract. At least one sentence in the abstract should mention NPI.

Response: The NPI information was added in the abstract.

Comment: The authors have responded to a query of an earlier reviewer that ‘gypsum was applied to improve marginal soil quality to support economic crop productivity.’ None of the soils in this study was sodic. Then gypsum would only have supplied Ca and SO42-. First of all, if these two were not deficient in the soil (whether sufficient or deficient in the soil was not mentioned in the article), what was the original intent to apply the gypsum. Also, I could not understand how gypsum could improve different labile C pools in soil by only supplying Ca and SO42- if they were not deficient to limit microbial activity. Observing the pH, I don’t think Ca would be deficient in these soils. S may be deficient I don’t know as there is no data on that. If gypsum is not amending any inherent problem like sodicity (which is not the case here as no soil was sodic) or supplying any nutrient which is deficient, then how it has a significant effect on improving soil quality and support economic crop productivity?

Response: While none of the soils were sodic, they were less productive soils, especially Marvyn loamy sand (Shorter, Alabama) and Omulga silt loam (Piketon, Ohio) soils were marginal or degraded soils. While the Marvyn loamy sand is a low water holding capacity, poor aggregate stability, and low SOM content marginal sandy soil, the Omulga silt loam is a compacted fragipan and low SOM content degraded soil. 

While most of our studied soil nutrient content data associated with FGD gypsum have already been published (Gonzalez JM, Dick WA, Islam KR, Watts DB, Fausey NR, Flanagan DC, VanToai TT, Batte MT, Reeder RC, Kost D, Shedekar VS. (2022) Gypsum and cereal rye cover crops affect soil chemistry: Trace metals and plant nutrients. Soil Science Society of America Journal. https://wileyonlinelibrary.com/journal/saj2), the exchangeable Ca and S concentration of all the soils were included in the manuscript, as per suggested.

FGD gypsum not only provides Ca and S, but it also provides Mg, K, and micronutrients. The cationic bridging effects of Ca promote flocculation of clay and SOC to form soil macroaggregate stability (>250 µm size aggregates) that lead to accumulated, macroaggregate protected organic C (POC) as particulate organic matter (POM), which is one of the labile C pools. Moreover, gypsum decreases surface crusts formation, acts as an electrolyte to facilitate water infiltration due to its higher solubility than lime, and reduce soluble reactive phosphorus via runoff and tile drainage. All these small but proactive effects of FGD gypsum are expected to improve microbial activity, influence SOC lability and accumulation, and affect crop productivity in conjunction with no-till and cropping diversity with cover crops. 

Comment: L 88-90: Mention the type of soil where such results were found

Response: As suggested, a brief description on soil type was added. “Recently, a few studies have reported that gypsum application favors soil biology and enzymatic activity in Hagerstown silt loam (pH 6.2) in Pennsylvania, USA [16] and in alkaline-saline clay soils in northwest China [24]. Gypsum amendments provide several nutrients essential for crop growth in Marvyn loamy sand (marginal soil with poor aggregate stability and low SOM content) in Alabama (Shorter), Blount silt loam in Indiana (Farmland), Hoytville clay in northwestern Ohio, and Omulga silt loam (degraded soil with compaction and low SOM content) in southern Ohio, respectively [21]”. 

Comment: L 99: Rewrite the sentence to avoid repetition of similar words

Response: Thanks. Revised accordingly.

Comment: L 189: Write ‘physical properties’ instead of ‘physical analysis’

Response: Thanks. Corrected, as per suggestion.

Comment: L 196: Put the starting parenthesis ‘(‘before ‘SMBC:SOC’)

Response: Edited accordingly.

Comment: L 217-218: Authors have mentioned that “The labile C pool was considered as a portion of the SOC that was comprised of the combined pools of active C, SMBC, and cold and hot-water C.” However, active C, SMBC, and cold- and hot- water C were not extracted sequentially from the same sample; they were extracted from separate samples with different methods. Each of them may a part of the labile C; but they are overlapping themselves. As they are not mutually exclusive, adding them up clearly overestimates the CL, and at the same time underestimates the non-labile C as the later was derived by subtracting CL from total SOC. Hence, the parameters derived from CL are also questionable. The authors should find a way to rectify this issue or give an appropriate rebuttal.

Response: Current statement about the “labile C pool” written in lines 217 - 218 has been corrected. The conceptually defined labile C pools such as active C, SMBC, and cold- and hot- water C were determined by KMnO4 oxidation, extraction with distilled deionized water, or K2SO4 solution using different methods. Separate soil samples were used to measure each C pool. It is not a combined C pool, each C pool was determined separately from SOC. That’s why we have different CL, CLI, and CMI values associated with active C, SMBC, and cold- and hot- water C pools, respectively (Table 1 to 5). 

Comment: Table 1, last row: This value of SMBC (1228 mg kg-1) seems too high, might be a typo, please check; If not a typo, then how come so huge difference is not significant?

Response: Thanks. It’s a typo. Corrected accordingly.

Comment: L 280-281: 2.2 kg or Mg ha-1?

Response: Corrected accordingly (2.2 Mg/ha)

Comment: L 317-318: The authors have seen some benefits of 2.2 Mg/ha gypsum application on SMBC, SMBC:SOC, and CMI; but, without the mention of its effect on crop yield, the information remains incomplete. They should at least include the average yield of four years under different treatments.

Response: Thank you for the comment. Detailed information associated with gypsum, crop rotation, and cover crop impact on crop yields was published in our earlier papers (Cited ref # 32 and 33 in the manuscript reference list). As the crop yield data were already published, we have tried to minimize the data overlapping and repetitions in incoming and future papers except soil quality indexing and evaluation. 

Cited ref. # 32: Batte MT, Dick WA, Fausey NR, Flanagan DC, Gonzalez JM, Islam R, Reeder R, VanToai T, Watts DB. (2018) Cover crops and gypsum applications: Soybean and corn yield and profitability impacts. American Society of Farm Manager and Rural Appraisers 8:47-71.

Cited ref. # 33: Raut Y, Shedekar V, Islam K, Gonzalez J, Watts D, Dick W, Flanagan D, Fausey N, Batte M, Reeder R, VanToai T. (2020) Soybean yield response to gypsum soil amendment, cover crop and rotation. Agricultural and Environmental Letters. https://doi.org/10.1002/ael2.20020.

---

## [Decision Letter · Decision Letter 2]

12 Sep 2022

Gypsum, crop rotation and cover crop impacts on soil organic carbon and biological dynamics in rainfed transitional no-till corn-soybean systems

PONE-D-22-07617R2

Dear Dr. Islam,

We’re pleased to inform you that your manuscript has been judged scientifically suitable for publication and will be formally accepted for publication once it meets all outstanding technical requirements.

Kind regards,

Debarati Bhaduri, Ph.D.

Academic Editor

PLOS ONE

Additional Editor Comments (optional):

All suggested changes have been suitably addressed.

Reviewers' comments:

Reviewer's Responses to Questions

**Comments to the Author**

1. If the authors have adequately addressed your comments raised in a previous round of review and you feel that this manuscript is now acceptable for publication, you may indicate that here to bypass the “Comments to the Author” section, enter your conflict of interest statement in the “Confidential to Editor” section, and submit your "Accept" recommendation.

Reviewer #2: All comments have been addressed

2. Is the manuscript technically sound, and do the data support the conclusions?

Reviewer #2: Yes

3. Has the statistical analysis been performed appropriately and rigorously? 

Reviewer #2: Yes

4. Have the authors made all data underlying the findings in their manuscript fully available?

Reviewer #2: Yes

5. Is the manuscript presented in an intelligible fashion and written in standard English?

Reviewer #2: Yes

6. Review Comments to the Author

Reviewer #2: The authors have tried to address the comments made by me. I have no further comments. The article in its present state may be accepted for publication.

7. PLOS authors have the option to publish the peer review history of their article (what does this mean?). If published, this will include your full peer review and any attached files.

Reviewer #2: No

---

## [Editor Report · Acceptance letter]

19 Sep 2022

PONE-D-22-07617R2 

Gypsum, crop rotation, and cover crop impacts on soil organic carbon and biological dynamics in rainfed transitional no-till corn-soybean systems 

Dear Dr. Islam:

I'm pleased to inform you that your manuscript has been deemed suitable for publication in PLOS ONE. Congratulations! Your manuscript is now with our production department. 

Kind regards, 

on behalf of

Dr. Debarati Bhaduri 

Academic Editor

PLOS ONE